# Nickel Catalysts on Carbon-Mineral Sapropel-Based Supports for Liquid-Phase Hydrogenation of Nitrobenzene

**Elena N. Terekhova** [1], **Olga B. Belskaya** [1,*] , **Rinat R. Izmaylov** [1], **Mikhail V. Trenikhin** [1] **and Vladimir A. Likholobov** [2]

[1]  Center of New Chemical Technologies, Boreskov Institute of Catalysis, 54 Neftezavodskaya Street, 644040 Omsk, Russia
[2]  Boreskov Institute of Catalysis, Siberian Branch, Russian Academy of Sciences, 5 Acad. Lavrentieva Ave., 630090 Novosibirsk, Russia
*  Correspondence: obelska@ihcp.ru; Tel.: +7-(3812)-670-474

**Abstract:** Nickel catalysts with carbon-mineral supports derived from sapropel were synthesized; the effect exerted by the nature of the support (type of the initial sapropel) and active component precursor on the activity of the catalysts in the model reaction of liquid-phase nitrobenzene hydrogenation was studied. The catalysts, synthesized using the support with a smaller fraction of carbon, were more active irrespective of the precursor nature. The highest activity was observed for the catalysts synthesized from nickel nitrate and formate; nitrobenzene conversion was 65% and 51%, respectively, after 1 h of reaction. The catalysts retained high activity after six reaction cycles at 100% aniline selectivity. The presence of sulfur in the nickel precursor deteriorated the catalytic activity (convection less than 3%) due to formation of the sulfide phase.

**Keywords:** sapropel; supported nickel catalyst; liquid-phase nitrobenzene hydrogenation

## 1. Introduction

Supported nickel catalysts as the components of mono- and bimetallic systems are marketable catalytic compositions due to their high activity and lower cost in comparison with catalysts based on the platinum group metals. They are widely used in hydrogenation reactions, for example, hydrogenation of various organic nitrocompounds to the corresponding amines [1–3], 5-hydroxymethylfurfural obtained from biomass to 2.5-bis(hydroxymethyl)furan and 2.5-dimethylfuran [4], acid hydrolyzates of lignocellulosic biomass to obtain xylite [5], as well as selective hydrogenation of arylhalogenide carbonyl compounds [6], biorenewable levulinic acid to valeric acid [7], and various aromatic compounds [8,9]. Nickel catalysts are employed not only in hydrogenation reactions but also in the reforming of methane and glycerol [10], reforming of oil tar, gasification of biomass, methanation and catalytic cracking of hydrocarbons, heavy crude oil refining [11], hydrodeoxygenation [12,13], and other reactions.

Supports of nickel catalysts are often represented by oxide compounds, such as oxides of aluminum [14], silicon [15], or cerium [16] as well as aluminosilicates [17,18]. Various carbon materials are also applied [1,5,19–22], particularly biocarbons derived from natural feedstock [20,21].

Earlier, we demonstrated that materials containing both the oxide and carbon parts could be used as catalyst supports [23–26]. Such carbon-mineral materials (CM) were derived from sapropel; their obvious advantages include high strength, which is caused by the presence of a mineral component, and high porosity provided by the carbon component. The low cost of the initial feedstock along with the possibility to control acid-base properties and textural characteristics by chemical treatment considerably extend the application field of porous CM produced from sapropels.

It should be acknowledged that sapropels, the bottom sediments of freshwater lakes, are complex systems with an organomineral composition. They contain up to 90% of organic matter, which includes humic, easy-to-hydrolyze, difficult-to-hydrolyze, and water-soluble organic substances, where the carbon fraction constitutes up to 69%, and also a wide spectrum of micro- and macroelements. Sapropels are the renewable, natural feedstock. Their classification is based on the content of mineral part (ash), according to which three main types are distinguished: organic sapropels (up to 30% of ash), organomineral (from 30 to 65% of ash), and mineral (65–88% of ash) ones [27]. The chemical composition of the initial sapropel is important for obtaining CM with the desired properties and should be taken into account in the synthesis of supported catalysts.

In our previous works [23,26,28], we established the influence of the nature of the initial sapropel (ratios between organic and mineral parts) on the properties of the carbon-mineral material obtained from it as well as the important role of the acid treatment of the support, which makes it possible to increase the specific surface area and the concentration of functional oxygen-containing groups. This modification led not only to an improvement in the cracking properties of catalysts with respect to large organic molecules [23,28] but also to a decrease in the size of supported nickel particles [23,26]. Another effective tool that made it possible to increase the dispersion of the active metal was the transition to the bimetallic (Ni–Mo and Ni–Cu) systems. With the use of such catalysts, with a decrease in the amount of nickel, an increase in activity was observed during the hydroconversion of the complex organic substance of sapropel to obtain bio-oil. It is important that the properties of catalysts supported on carbon-mineral materials from sapropel were comparable to those of nickel-containing catalysts using more expensive synthetic supports such as alumina, amorphous aluminosilicate, and ultrastable zeolite [28].

In this work, we considered the possibility of controlling the hydrogenating activity of nickel without using an additional metal but by varying the nature of the active component precursor. Additionally, in this study, the issue of the stability of Ni/CM catalysts based on sapropel and the possibility of their regeneration was considered; multi-cycle experiments were performed with the control of the nickel content in the catalysts. To obtain supports, we also used sapropels from other deposits in order to show that the developed approaches to the synthesis and modification of composite carbon-mineral supports are applicable to any materials of this type, regardless of the raw material localization.

## 2. Results and Discussion

### 2.1. Investigation of CM Supports

In this study, the objects of investigation were represented by two carbon-mineral materials: CM-M and CM-O; they were produced, respectively, from mineral and organic sapropels, which differed in the ash content and chemical composition [29,30]. According to XRD, CM-M and CM-O had close composition of the mineral part, which was represented by $SiO_2$ quartz as the main phase; in addition, muscovite $KAl_2(Si_3Al)O_{10}(OH)_2$, albite $(Na_{0.84}Ca_{0.16})Al_{1.16}Si_{2.84}O_8$, microcline $(K_{0.95}Na_{0.05})AlSi_3O_8$, calcium aluminosilicate $Ca_{0.88}Al_{1.77}Si_{2.2}O_8$, and other substances were detected [30]. However, the content of the mineral part differed significantly, which was reflected in different values of the bulk density and ash content for the corresponding CM (Table 1).

**Table 1.** Main physicochemical characteristics of CM and their estimated surface acidity.

| Sample | $\rho$, g·cm$^{-3}$ | $A^{daf}$, wt.% | $V_{pore}^{H_2O}$, cm$^3$·g$^{-1}$ | pH$_{PZC}$ | [O$^=$], mmol·g$^{-1}$ * |
|---|---|---|---|---|---|
| CM-M | 0.65 | 79.7 | 0.17 | 2.6 | 0.24 |
| CM-O | 0.49 | 46.2 | 0.15 | 2.4 | 0.31 |

\* The total amount of oxygen-containing surface groups.

The investigation of the textural characteristics of CM-M and CM-O revealed (Table 2) that, despite differences in the composition of the native CM precursor, carbonization of sapropel followed by acidic treatment can lead to supports with close specific surface areas

and pore volumes. The analysis of the pore size distribution also showed a similar pattern. Overall, the produced materials are the macroporous objects with the total content of meso- and macropores above 65%. However, a greater contribution of micropores in the case of CM-O resulted in a somewhat higher specific surface area at a lower pore volume. In addition, the analysis of the surface acidic properties, which was performed by measuring the PZC and estimating the total amount of oxygen-containing (carbonyl, phenol, and carboxyl) groups on the surface (Table 1), demonstrated that both materials had close acidic properties and could be considered as the acidic supports.

**Table 2.** Textural characteristics of CM.

| Sample | $S_{BET}$, m$^2$·g$^{-1}$ | $V_{pore}$, cm$^3$·g$^{-1}$ | Pore Fraction, % | | |
| | | | Micropores (<2 nm) | Mesopores (2–50 nm) | Macropores (50–600 nm) |
|---|---|---|---|---|---|
| CM-M | 161 | 0.25 | 27 | 42 | 31 |
| CM-O | 172 | 0.19 | 34 | 33 | 33 |

Thus, the chosen synthesis and pretreatment methods made it possible to produce carbon-mineral materials with close textural and acid-base properties at different chemical compositions. This will allow revealing the effect of the nature of support precursors (differences in the ratio of mineral and organic parts in the sapropels used in the study) on the anchoring of an active component and hydrogenation activity of supported nickel catalysts.

## 2.2. Synthesis and Investigation of Ni-Containing Catalysts

The catalysts under consideration were synthesized by incipient wetness impregnation of the supports, which were produced from different sapropels (CM-M and CM-O), with saturated solutions of nickel salts. The calculated content of nickel was 10 wt.%; its actual content, estimated by atomic absorption spectrometry (AAS) after calcination and dissolution of the samples, is shown in Table 3. The revealed differences in the nickel content, which may be associated with losses of the active component due to the complex multistep impregnation procedure, was taken into account when we compared the catalytic activity. The general elemental composition of the catalysts is also illustrated in Figure S1 by the example of the EDX analysis of a section of the "Ni/CM-M, formate" sample.

**Table 3.** Ni content in the samples as determined by AAS.

| Precursor | Ni Content, wt.% | |
| | CM-M | CM-O |
|---|---|---|
| $Ni(NO_3)_2$ | 10.4 | 10.6 |
| $Ni(CH_3CO_2)_2$ | 9.1 | 10.3 |
| $(NH_4)_2Ni(SO_4)_2$ | 9.2 | 10.2 |
| $Ni(CHO_2)_2$ | 9.9 | 9.4 |

The formation conditions of metal particles during the synthesis of supported metal catalysts are conventionally determined by the temperature-programmed reduction. This method was employed to examine the samples with the deposited nickel precursor after high-temperature calcination (Figure 1 and Table 4). The experiment was carried out up to 600 °C because, above this temperature, an intense methanation is observed (Figure S2), which accelerates in the presence of deposited metal, thus deteriorating the TPR profiles and making their interpretation impossible. A preliminary study of CM without deposited metal under the TPR conditions showed (Figure S2) that the contribution of the reduction in metals present in the initial sapropel was insignificant and could be neglected in the analysis of TPR data. Although the reduction in nickel is usually performed at a temperature of ca. 600–650 °C [31], the TPR profiles may contain the low-temperature hydrogen consumption

peaks (up to 400 °C), which are associated with the NiO → Ni transition. The shape and intensity of such peaks depend on the nature of both the CM support and the precursor of active metal. The most intense peaks correspond to the reduction in nickel deposited on CM-M, although, in the case of nitrate and formate, the fraction of "low-temperature" nickel is also high during its deposition on CM-O. In addition, the presence of steps on the consumption peak (for example, at 180 and 220 °C) testifies to inhomogeneity of the reduction process, which may be related to the size and/or localization of NiO particles. In the case of CM-O containing a large fraction of carbon, an increase in hydrogen consumption in the high-temperature region was observed for all the samples (especially for that produced from sulfate), which was caused by the conversion of carbon (methanation). The quantitative analysis of the TPR data demonstrated that, at a stoichiometric ratio $nH_2/nNiO$ equal to 1, such ratios in the case of nickel reduction on CM-M did not exceed 0.3. Their higher values in the case of CM-O were caused by the hydrogen consumption for its interaction with the carbon part of the support. It was difficult to estimate the contribution of "low-temperature" nickel to the properties of final catalysts; however, differences in the TPR profiles reflected the existing differences in the formation of deposited nickel particles depending on the composition of the support and the nature of the active metal precursor.

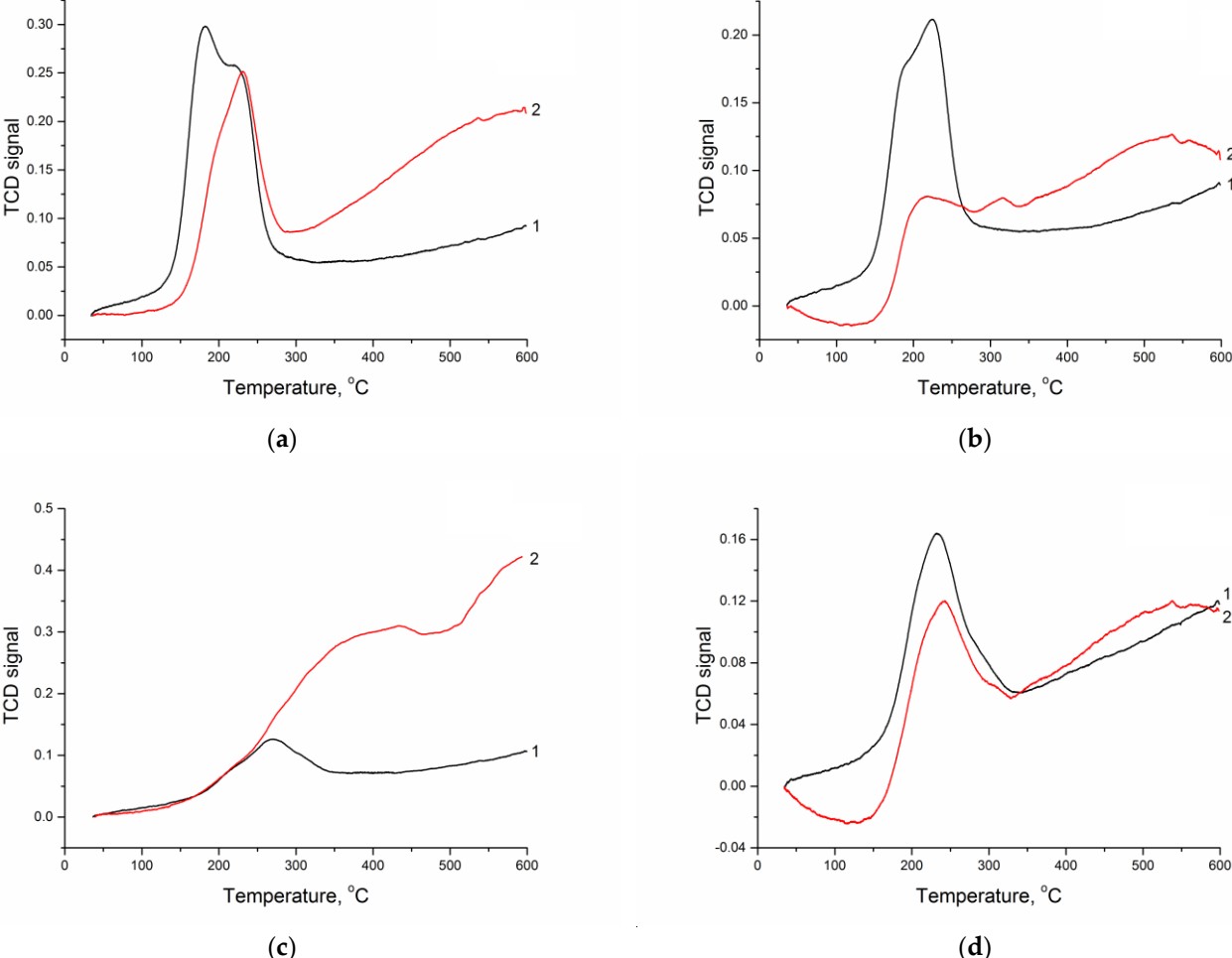

**Figure 1.** TPR profiles of nickel oxide species on the surface of (1) CM-M and (2) CM-O (nickel precursors: (**a**) nitrate, (**b**) acetate, (**c**) sulfate, and (**d**) formate).

**Table 4.** Ratios of the amount of consumed hydrogen and nickel content according to TPR and CSR of nickel in the catalysts reduced at 650 °C according to XRD in dependence of the nature of support and precursor of the active component.

| Precursor | Catalyst | nH₂/nNiO | CSR, nm (111) |
|:---:|:---:|:---:|:---:|
| $Ni(NO_3)_2$ | Ni/CM-M | 0.2 | 20.0 |
|  | Ni/CM-O | 0.4 | 28.2 |
| $Ni(CH_3CO_2)_2$ | Ni/CM-M | 0.3 | 31.3 |
|  | Ni/CM-O | 0.4 | 31.0 |
| $(NH_4)_2Ni(SO_4)_2$ | Ni/CM-M | 0.2 | 31.7 |
|  | Ni/CM-O | 1.0 | 34.5 |
| $Ni(CHO_2)_2$ | Ni/CM-M | 0.2 | 20.9 |
|  | Ni/CM-O | 0.5 | 47.3 |

To increase the degree of nickel reduction before catalytic experiments, the reduction temperature was increased to the conventional 650 °C with holding in a hydrogen flow for 4 h. The formation of nickel particles as a result of such treatment was confirmed by the XRD data; the presented diffraction patterns (Figure 2) contained reflections that were typical of the metallic nickel (Figure S3) phase with cubic syngony in the absence of signals corresponding to nickel oxide. The other reflections on the diffraction patterns were associated with components of the mineral matrix of supports (Figure S4). It should be noted that the use of nickel sulfate as a precursor resulted in the formation of the nickel sulfide phase (2θ = 55.16°) (Figure 2, line 3).

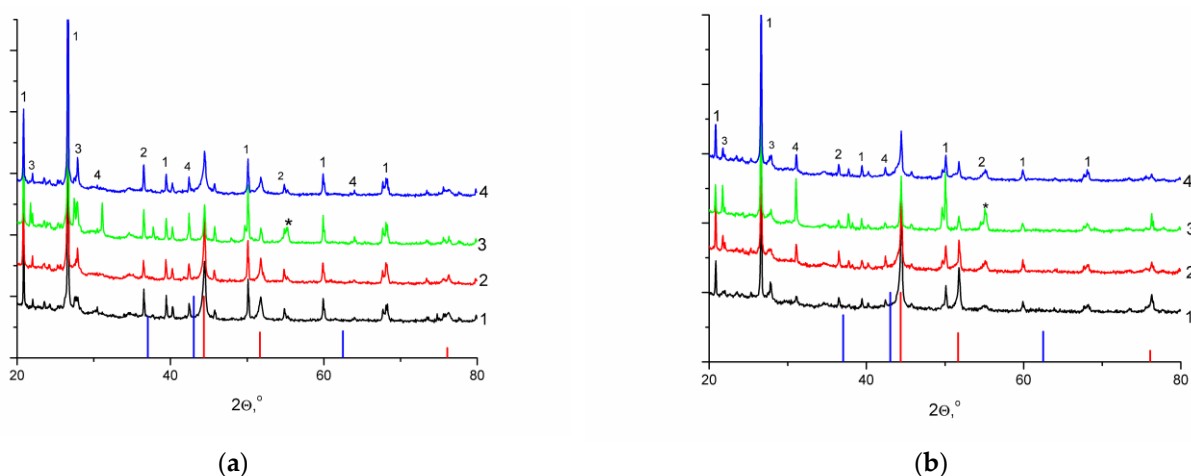

(**a**)                                                                            (**b**)

**Figure 2.** Diffraction patterns of the reduced catalyst samples, (**a**) on CM-M, (**b**) on CM-O; precursors: (1) nitrate, (2) acetate, (3) sulfate, (4) formate (red line: basic reflections of Ni, PDF #03-065-380; blue line: basic reflections of NiO, PDF #01-071-4750; * basic reflections of heazlewoodite, Ni₃S₂, PDF #01-085-1802; (1) SiO₂, PDF #01-070-3755; (2) muscovite KAl₂(Si₃Al)O₁₀(OH)₂, PDF #00-007-0025; (3) albite (Na₀.₈₄Ca₀.₁₆)Al₁.₁₆Si₂.₈₄O₈, PDF #01-076-0927; (4) microcline (K₀.₉₅Na₀.₀₅)AlSi₃O₈, PDF #01-084-1455).

A comparison of nickel particle sizes for the series of samples reduced at 650 °C was made by calculating the CSR values from the corresponding diffraction peaks (Table 4). The deposition of nickel on CM-M resulted in the formation of particles with a smaller size. The minimum CSR values were obtained for the Ni/CM-M samples that were synthesized using nickel nitrate and formate. This result was confirmed by the electron microscopy data. It followed, from Figure 3, that spherical nickel particles were uniformly distributed in these samples and had a size of 10–12 nm ( Figures 3a–f and S5). When CM-O was used, both the smaller particles and the regions of agglomerated particles were observed (Figure 3g–i).

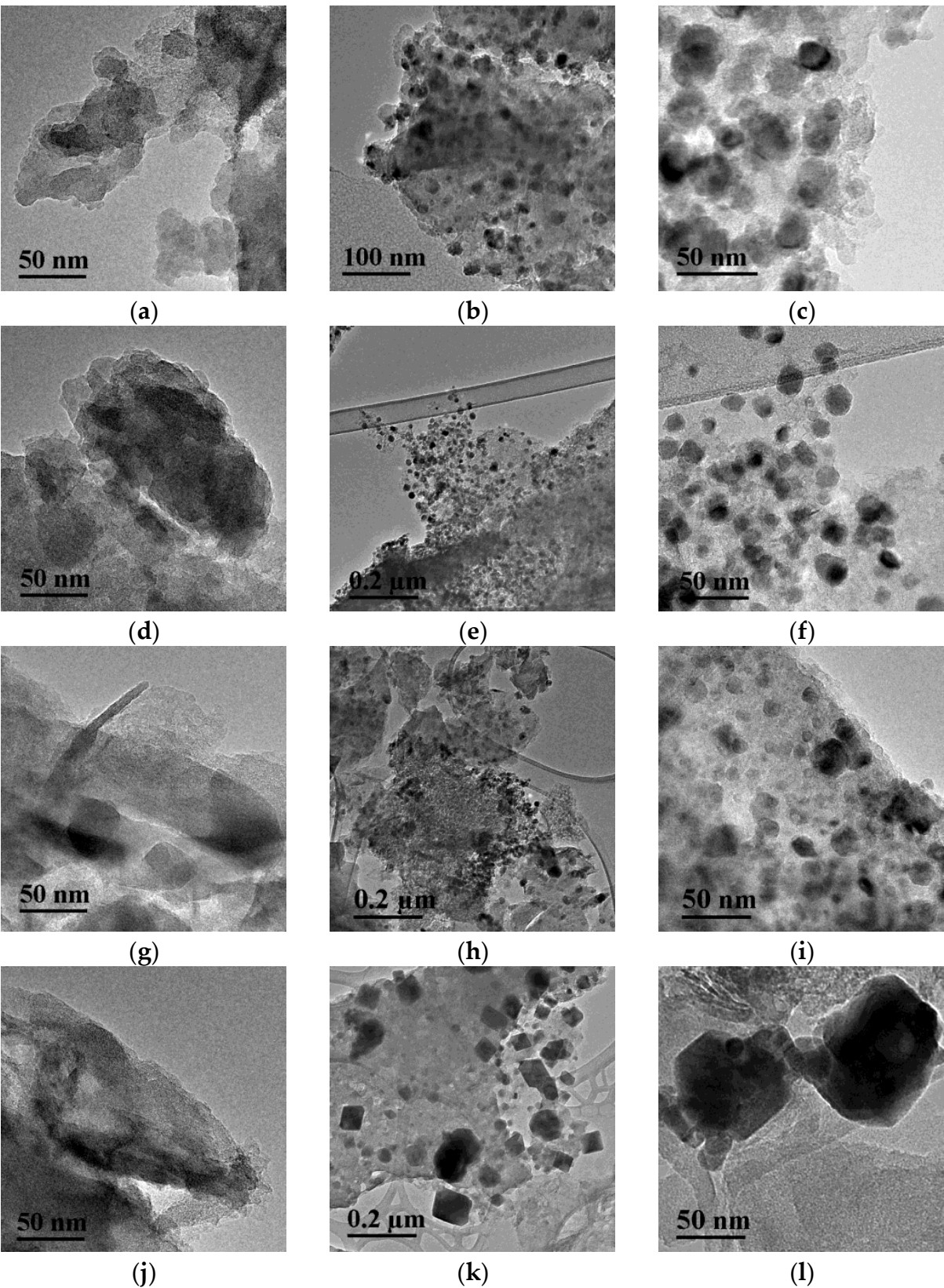

**Figure 3.** TEM images of supports ((**a**,**d**) CM-M, (**g**,**j**) CM-O) and catalysts ((**b**,**c**) Ni/CM-M, nitrate; (**e**,**f**) Ni/CM-M, formate; (**h**,**i**) Ni/CM-O, formate; and (**k**,**l**) Ni/CM-O, sulfate) at different magnifications.

When nickel sulfate was used, at the average particle sizes of nickel close to other samples (according to XRD data), large particles of the "cubic" form were observed on the surface, which were assigned to the nickel sulfide particles, the phase of which was identified by XRD (Figure 3k,l). Although HRTEM could not characterize the structure of such particles, the analysis of EDS data (Figure 4) showed that nickel and sulfur were located at similar sites, thus verifying the possibility of their interaction.

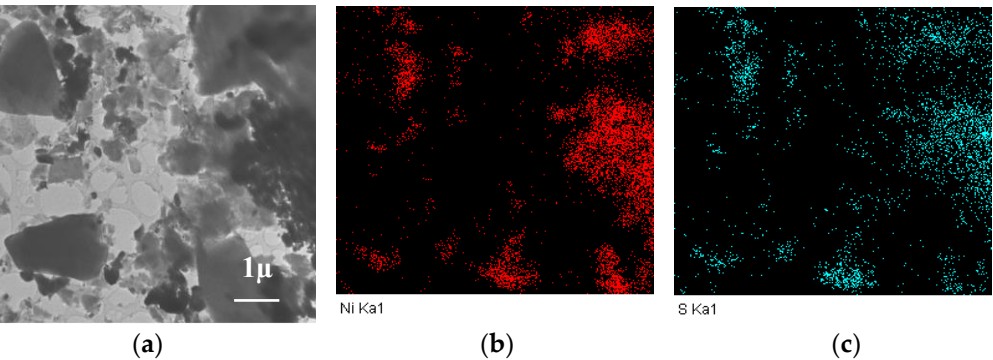

| (a) | (b) | (c) |

**Figure 4.** TEM image (**a**) and the distribution of nickel (**b**) and sulfur (**c**) according to EDS data for the Ni/CM-O catalyst synthesized using nickel sulfate.

### 2.3. Investigation of the Catalytic Activity of Ni/CM in Hydrogenation of Benzene

The liquid-phase hydrogenation of nitrobenzene is a well-known reaction, and many efficient catalytic compositions are used to carry it out. Palladium is often chosen as the active component, and expensive synthetic oxide and carbon materials are used as supports (Table S1). In some studies, this reaction (as in this work) is considered as a test for comparing the hydrogenation activity of a number of catalysts when optimizing their composition or synthesis conditions; in this case, mild conditions (low reaction temperature, short reaction time) were used to avoid reaching full conversion.

The catalytic activity of deposited nickel in dependence of the composition of support and the nature of precursor was estimated in the liquid-phase hydrogenation of nitrobenzene from the amount of hydrogen absorbed for 1 h and the absorption rate in the initial step of the reaction. Before testing, the samples were calcined in Ar and reduced at 650 °C (this temperature was justified, since, at a reduction temperature of 350 °C, corresponding to the production of "low-temperature" nickel, three times less active catalysts were formed (Figures S6 and S7)). Figure 5 displays the examples of such dependences for the catalysts synthesized with nickel nitrate and formate as the precursors. The only product of the reaction was aniline (the reaction products were identified with the use of NMR spectroscopy (Figure S8)).

The data presented in Figure 5 and Table 5 demonstrate essential differences between catalysts. Samples synthesized with the support made of the mineral-type sapropel, CM-M, showed a higher catalytic activity compared to the samples supported on CM-O, which was indicated by higher values of conversion and specific activity. Amorphous carbon, which dominated in the composition of CM-O, was quite active and was able not only to interact with hydrogen at the nickel reduction step but may partially have shielded the surface of nickel particles upon anchoring of the precursor on the carbon surface. In addition, there were data on the direct interaction of carbon with nickel, which could also have affected the hydrogenation activity of the metal [32,33].

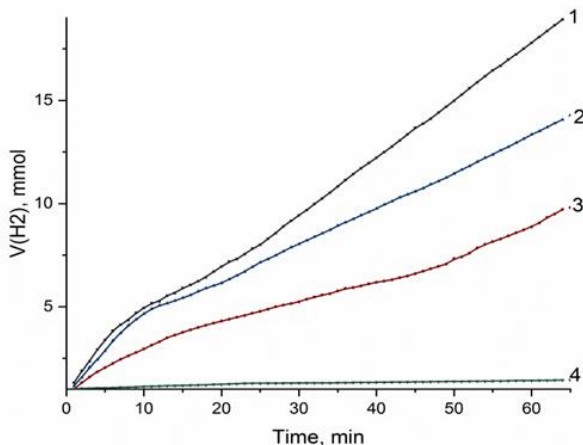

**Figure 5.** Hydrogen absorption curves for hydrogenation of nitrobenzene on nickel catalysts based on different types of sapropels ((1) Ni/CM-M, nickel nitrate as a precursor; (2) Ni/CM-M, nickel formate as a precursor; (3) Ni/CM-O, nickel nitrate as a precursor; and (4) Ni/CM-O, nickel formate as a precursor).

**Table 5.** Conversion of nitrobenzene (X, %) and specific catalytic activity of the tested catalysts (SCA, $mmol(H_2) \cdot g(Ni)^{-1} \cdot min^{-1}$).

|  | Support | $Ni(NO_3)_2$ | $Ni(CH_3CO_2)_2$ | $(NH_4)_2Ni(SO_4)_2$ | $Ni(CHO_2)_2$ |
|---|---|---|---|---|---|
| X | CM-M | 64.9 | 33.0 | 2.8 | 50.5 |
| | CM-O | 31.3 | 13.7 | 1.2 | 5.0 |
| SCA | CM-M | 0.7 | 0.6 | 0.1 | 0.7 |
| | CM-O | 0.3 | 0.1 | <<0.1 | 0.1 |

As for the precursors used in the study, most active were the catalysts supported on CM-M, which were synthesized from nickel nitrate and formate. The use of nickel nitrate allowed us to obtain the most active sample also in the CM-O series, probably due to the formation of more dispersed metal particles from this precursor (Table 4). The catalysts synthesized from nickel-ammonium sulfate showed a very low activity irrespective of the support nature. This may have been caused by the deactivation of a considerable nickel fraction due to the formation of $Ni_3S_2$ during high-temperature treatments.

Thus, CM-M is a more promising support for the catalyst synthesis by the chosen method. Presumably, the presence of a greater fraction of mineral components, primarily silicon oxide, promotes the preferential anchoring of the metal on these fragments and the formation of dispersed and active particles of deposited nickel. This assumption is consistent with the EDS data obtained for the Ni/CM-M sample. As follows from Figure 6, the locations of nickel most often corresponded to the regions with a high content of silicon.

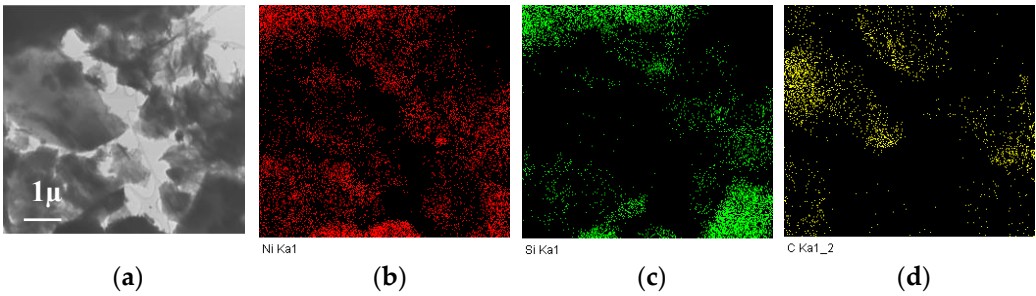

(**a**)　　　　　　　(**b**)　　　　　　　(**c**)　　　　　　　(**d**)

**Figure 6.** TEM image (**a**) and the distribution of nickel (**b**), silicon (**c**) and carbon (**d**) for the Ni/CM-M catalyst synthesized using nickel formate.

To elucidate the role of mineral and carbon components (in the carbon-mineral materials derived from natural feedstock) in the formation of the catalyst properties, model samples were synthesized using supports made of silicon and aluminum oxides (the highest content of these oxides was in the mineral part of CM (Figure S4)) and carbon. Carbon black of P-514 grade served as the carbon support. Some of the chosen supports had specific surface areas close to those for CM: $S_{BET}(\gamma\text{-}Al_2O_3, Condea) = 202 \text{ m}^2\cdot\text{g}^{-1}$, $S_{BET}(SiO_2, Salavat) = 333 \text{ m}^2\cdot\text{g}^{-1}$, and $S_{BET}(P\text{-}514, Omsk Carbon group) = 235 \text{ m}^2\cdot\text{g}^{-1}$. After the deposition of nickel (nickel nitrate as a precursor), high-temperature calcination and reduction reference samples were tested in hydrogenation of nitrobenzene. On $Ni/Al_2O_3$ and $Ni/SiO_2$ samples, nitrobenzene was completely converted during 1 h of the experiment, whereas, on the $Ni/P\text{-}514$ sample, its conversion reached only 4% under the same conditions. This experiment confirms the negative effect of carbon in CM on the hydrogenation activity of nickel under the chosen reaction conditions and allows substantiating the choice of the CM-M support for the synthesis of catalysts.

The most active Ni/CM-M catalysts, which were synthesized using nickel nitrate and formate as the precursors of the active component, were studied in a multi-cycle experiment. After each cycle (hydrogenation for 1 h at a hydrogen pressure of 2 MPa and temperature of 90 °C), the catalyst was separated from the reaction products by filtering and then dried, reduced, and used for hydrogenation of a fresh portion of nitrobenzene. It follows from Figure 7 that the conversion of nitrobenzene remained high during the entire experiment. The observed decrease in activity may be attributed to both the inevitable losses of the catalyst during filtering and the partial removal of nickel from the support surface. Thus, the analysis of the samples after six cycles of the experiment revealed a decrease in the nickel content in the catalysts by ca. 1 wt.%: from 10.4 to 9.4% when nickel nitrate served as the precursor and from 9.9 to 8.5% in the case of formate. This effect was related to a high metal content and, hence, a weak metal–support interaction and can be diminished by optimizing the nickel content and the methods of catalyst synthesis.

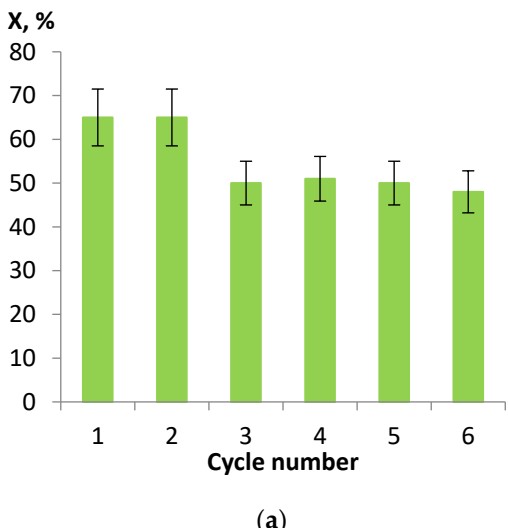

(**a**)

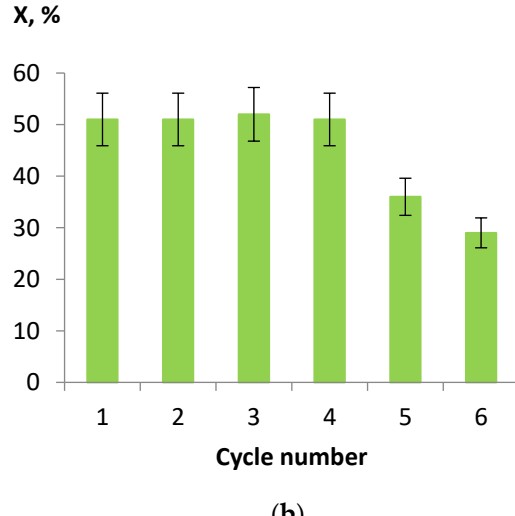

(**b**)

**Figure 7.** Results of multi-cycle experiments for nickel catalysts on CM-M, which were synthesized from nitrate (**a**) and formate (**b**) as the precursors, in hydrogenation of nitrobenzene (the reaction conditions: hydrogen pressure 2 MPa, temperature 90 °C, and time 60 min).

### 3. Materials and Methods

Carbon-mineral materials CM-O and CM-M, which served in this study as the catalyst supports, were prepared using organic and mineral sapropels from deposits in the Omsk region (Russia): Gorchakovskoye and Gorkoye lakes, correspondently. The carbonization of native sapropels was carried out in an argon medium at 600 °C for 30 min at a heating rate of 5 °C·min$^{-1}$, while the subsequent acidic treatment produced CM with the use of a

5 M $HNO_3$ aqueous solution according to [29]. The scheme of synthesis of carbon-mineral supports is additionally shown in Figure S9.

The bulk density of the CM samples ($\rho$) was assessed according to the state standard of Russia GOST 16190–70 "Sorbents. Methods for estimating of the bulk density". The essence of the method consists of the determination of the sorbent weight corresponding to a certain volume at the normalized compaction.

Ash content ($A^{daf}$) was estimated according to GOST 11022–95 "Solid mineral fuel. Methods for estimating the ash content". A sample was burned in a muffle furnace, which was heated to a temperature of $815 \pm 10$ °C and held at this temperature until a constant weight was achieved. The ash content percentage was calculated from the weight of a residue after calcination.

The total pore volume with respect to water ($V_{pore}^{H_2O}$) was found according to GOST 17219-71 "Active carbons. A method for estimating the total pore volume with respect to water". The method is based on filling the 0.5–104 nm pores with water upon boiling of a sample in water and the removal of the excess water from the grain surface by evacuation.

The textural characteristics of the samples were investigated by low-temperature nitrogen adsorption on a Sorptomatic 1900 Carlo Erba instrument (Milan, Italy). Prior to the adsorption measurements, the samples were evacuated at 300 °C for 6 h to a residual pressure not higher than 0.1 Pa. Specific surface area of the samples was calculated according to BET from the adsorption isotherm in the range of equilibrium relative pressures of nitrogen $P/P_0 = 0.05$–0.25. The micropore volume was estimated using a comparative t-method. The mesopore volume was calculated from the difference between pore volume at the equilibrium relative pressure of nitrogen vapor $P/P_0 = 0.990$ and micropore volume. The macropore volume was found from the difference between pore volumes measured at $P/P_0 = 0.999$ and $P/P_0 = 0.990$.

To assess the acid-base surface properties of the produced CM, points of zero charge ($pH_{PZC}$) were determined by the method proposed in [34]. To find the PZC, carbon-mineral materials were placed in aqueous solutions with different initial pH values. Solutions with the initial pH value ranging from 1 to 13 were prepared using HCl and NaOH. After establishing a constant pH value, the measurements were repeated. The $pH_{PZC}$ value corresponded to pH on a plateau of the dependence of final (equilibrium) pH on the initial pH. The pH measurements were performed using a Seven Multi (Mettler Toledo, Greifensee, Switzerland) ionometer with a combined pH electrode and a solid reference electrolyte XEROLYT.

The total amount of surface acid groups was determined by selective neutralization according to the method proposed in [35].

X-ray diffraction analysis of the samples was carried on a D8 Advance (Bruker, Ettlingen, Germany) X-ray powder diffractometer with a CuK$\alpha$ source ($\lambda = 0.15406$ nm). The following measurement modes were used: scanning step 0.050°, signal accumulation time 2 s·point$^{-1}$, voltage 40 kV, filament current 40 mA, and scanning range 5–80°. The acquired diffraction patterns were interpreted using the ICDD PDF-2 powder diffraction database. Coherent scattering regions (CSR) were calculated by the Scherer equation.

Catalysts with the specified nickel content of 10 wt.% were synthesized by incipient wetness impregnation of the supports with solutions of nickel salts. The number of impregnations depended on the solubility of the salts used in the study. Nickel precursors were represented by the salts of organic and mineral acids: nitrate $Ni(NO_3)_2·6H_2O$, acetate $Ni(CH_3CO_2)_2·4H_2O$, formate $Ni(CHO_2)_2·2H_2O$, and nickel-ammonium sulfate $(NH_4)_2Ni(SO_4)_2·6H_2O$. To transform the active metal into the oxide form, the samples were calcined in an argon medium at 650 °C for 16 h. Before catalytic testing, the catalysts were reduced at 650 °C for 4 h.

The chemical composition of the samples under consideration was determined on an AA-6300 Shimadzu (Kyoto, Japan) atomic absorption spectrometer. Nickel content was measured by atomic absorption in an air–acetylene flame (15.0—1.8 L/min) at a wavelength of 232.0 nm and a slit width of 0.2 nm.

Temperature-programmed reduction (TPR) of the tested samples (after calcination in argon medium at 650 °C for 16 h) was performed on an AutoChem II 2920 (Micromeritics, Norcross, GA, USA) chemisorption analyzer equipped with a highly sensitive thermal conductivity detector. The reduction was carried out using a calibrated mixture of 10 vol.% $H_2$ in argon. The temperature range was 35–600 °C. A measuring cell with the sample was heated at a rate of 10 °C·min$^{-1}$. A flow rate of the gas mixture through the reactor with the sample was 30 cm$^3$ (STP)·min$^{-1}$. The sample was held at 600 °C for 30 min in order to obtain the spectrum baseline.

A TEM study of the samples was carried out on a JEM-2100 (JEOL, Kyoto, Japan) transmission electron microscope (accelerating voltage 200 kV, crystal lattice resolution of the gold single crystal 0.145 nm).

The hydrogenation experiments were performed using a 1% solution of nitrobenzene. Ethanol (99 wt.%) served as a solvent. The reaction was performed at a pressure of 2 MPa and temperature 90 °C for 60 min on a laboratory setup consisting of an autoclave with a magnetic stirrer. The weight of the catalyst sample was 1.0 g, and the stirring rate was 1300 rpm$^{-1}$. The amount of absorbed hydrogen was recorded each minute using an EL-FLOW Select (Bronkhorst High-Tech, Ruurlo, The Netherlands) flow meter. Experimental data were presented as the amount of absorbed hydrogen (mol) versus time. Specific catalytic activity (SCA, mmol·g(Ni)$^{-1}$·min$^{-1}$) was calculated as a ratio of the hydrogen absorption rate to the weight of the catalyst active component (Ni). The rate was found from the slope of the curve in the initial rectilinear region of the graphical dependence of the absorbed hydrogen amount on time.

The reaction products were identified on an Avance-400 (Bruker, Billerica, MA, USA) NMR spectrometer in standard ampoules, having a diameter of 5 mm, at a frequency of 400 MHz ($^1$H) or 100.6 MHz ($^{13}$C) and temperature 25 °C in the pulsed mode. A signal from residual protons ($\delta$H 3.30 ppm, quintet) or carbon atoms $CD_3$ ($\delta$C 49.0 ppm, septet) in methanol-d4 served as the internal standard. The NMR spectra were simulated using the ACD/Labs 6.00 (Advanced Chemistry Development Inc., Toronto, ON, Canada) software package.

The quantitative composition of the reaction mixture was determined on a Chromos GC-1000 (Dzerzhinsk, Russia) gas chromatograph. The obtained chromatograms were processed using the Chromos software. Chromatographic analysis was performed on a capillary column DB-1 (dimethylsiloxane as the stationary liquid phase; L = 100 m; d = 0.25 mm) with a flame ionization detector. Concentrations were calculated using the internal normalization method. The conversion of nitrobenzene X (%) was calculated by the equation

$$X = (S_a / \sum S) \times 100\%,$$

where $S_a$ is the area of aniline peak on a chromatogram and $\sum S$ is the total area of peaks corresponding to the reaction components.

## 4. Conclusions

Nickel catalysts on carbon-mineral supports made of different types of sapropels were synthesized using nickel nitrate, acetate, and formate as well as nickel-ammonium sulfate as precursors of the active component; the synthesized catalysts were investigated in the liquid-phase hydrogenation of nitrobenzene.

The catalysts on the supports produced from the mineral-type sapropel showed a higher hydrogenation activity compared to those prepared from organic sapropel, irrespective of the nature of the active component precursor. The maximum activity was observed for the catalysts on CM-M, which were synthesized using nickel nitrate and formate; nitrobenzene conversion was 65% and 51%, respectively, after 1 h of reaction. These catalysts were able to retain their activity during several cycles of operation. The observed decrease in activity (decrease in conversion did not exceed 20%) were attributed to both the inevitable losses of the catalyst during filtering and the partial removal of nickel from the support surface.

Although in the chosen reaction the presence of a considerable carbon amount in CM-O led to a detrimental lowering of the catalytic activity, the possibility of controlling the hydrogenation properties of nickel by implementing the Ni-C interaction may be useful for selective hydrogenation of polyfunctional compounds [32].

**Supplementary Materials:** The following supporting information can be downloaded at https://www.mdpi.com/article/10.3390/catal13010082/s1. Figure S1: EDX spectra and atomic % of elements in sample Ni/CM-M, formate. Figure S2: TPR profile of CMM support. Figure S3: Diffraction patterns of Ni/CM-M sample reduced at 600 °C with $Ni(NO_3)_2$ as a precursor. Reflections associated with the nickel metal phase are noted. Figure S4: Diffraction patterns of CM-M and CM-O samples ((1) $SiO_2$, PDF #01-070-3755; (2) muscovite $KAl_2(Si_3Al)O_{10}(OH)_2$, PDF #00-007-0025; (3) albite $(Na_{0.84}Ca_{0.16})Al_{1.16}Si_{2.84}O_8$, PDF #01-076-0927; (4) microcline $(K_{0.95}Na_{0.05})AlSi_3O_8$, PDF #01-084-1455). Figure S5: Size distribution of nickel particles from TEM data: (a) Ni/CM-M, nitrate; (b) Ni/CM-M, formate. Table S1: The comparison of the studied catalysts' activity with other catalysts in the reaction of liquid-phase nitrobenzene hydrogenation. Figure S6: Diffraction patterns of Ni/CM-M sample reduced at 350 °C (red line: basic reflections of $NiO_2$, PDF #01-085-1977; black line: basic reflections of Ni, PDF #03-065-380; blue line: basic reflections of NiO, PDF #01-071-4750). Figure S7: Hydrogen absorption curves for hydrogenation of nitrobenzene on Ni/CM-M with nickel nitrate as a precursor, reduced at: (1) 350 °C, (2) 650 °C (the reaction conditions: hydrogen pressure 2 MPa, temperature 90 °C, and time 60 min). Figure S8: NMR ($^1$H) spectra of nitrobenzene hydrogenation reaction products using nickel catalysts based on sapropels of various natures with $Ni(NO_3)_2$ as a precursor ((a) Ni/CM-M, (b) Ni/CM-O). Figure S9: Scheme for the synthesis of CM support from sapropel.

**Author Contributions:** Conceptualization, E.N.T., O.B.B. and V.A.L.; methodology, E.N.T.; validation, O.B.B. and V.A.L.; formal analysis, R.R.I. and M.V.T.; investigation, E.N.T.; data curation, E.N.T., R.R.I. and M.V.T.; writing—original draft preparation, E.N.T.; writing—review and editing, O.B.B. and M.V.T.; visualization, O.B.B. and V.A.L.; supervision, O.B.B. and V.A.L.; project administration, O.B.B. and V.A.L. All authors have read and agreed to the published version of the manuscript.

**Funding:** This work was supported by the Ministry of Science and Higher Education of the Russian Federation within the governmental order for Boreskov Institute of Catalysis (project AAAA-A21-121011490008-3).

**Data Availability Statement:** Data are contained in the article and Supplementary Materials. Any additional data are available on request from the corresponding author.

**Acknowledgments:** The authors thank Aleksandra A. Alekseeva for her help in conducting experiments and Ivan V. Muromtsev, Galina G. Savelyeva, Tatiana I. Gulyaeva, and Sergey N. Evdokimov for their participation in the study of the samples. The research was performed using equipment of the Shared-Use Center "National Center for the Study of Catalysts" at the Boreskov Institute of Catalysis and the Omsk Regional Center of Collective Usage, Siberian Branch of the Russian Academy of Sciences.

**Conflicts of Interest:** The authors declare no conflict of interest.

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
