# Peer review of "Nickel Catalysts on Carbon-Mineral Sapropel-Based Supports for Liquid-Phase Hydrogenation of Nitrobenzene"

_catalysts, doi:10.3390/catal13010082_

Round 1

Reviewer 1 Report

There are a few issues to be addressed in this paper. Abstract and conclusion sections need to quantitatively summarize the findings of the work. Page 2, surface area data have too many significant figures. P3, L114, take out it should be noted that. Figure 2, index the peaks. P5, L160, take out it is seen that. P8, L265, take out It should be noted that. Figure 7, give error bars on all data points.  

Author Response

Our general response:

The authors thank the referee for analyzing the presented material. We fully agree with your specific comments and hope that changing the manuscript, taking into account all the comments, will bring the article closer to the level of the Journal. All changes are highlighted in yellow.

Abstract and conclusion sections need to quantitatively summarize the findings of the work.

Factual data was added to the Abstract and Conclusion.

Page 2, surface area data have too many significant figures.

Changes made to table 2.

P3, L114, take out it should be noted that. P5, L160, take out it is seen that. P8, L265, take out It should be noted that.

Changes made on pages 3, 5 and 8.

Figure 2, index the peaks

We have labeled the peaks in Fig. 2, and their assignment to the phases of nickel, nickel oxide, and compounds present in the support is given in the caption to the figure. In addition, the detailed identification of the metallic nickel phase is shown in Fig. S3.

Figure 7, give error bars on all data points.

The necessary changes are made in Fig. 7.

Reviewer 2 Report

The manuscript entitled “Nickel Catalysts on Carbon-Mineral Sapropel-Based Supports for Liquid-Phase Hydrogenation of Nitrobenzene” describes Preparation of nickel catalysts with the carbon-mineral supports derived from sapropel and has been used as catalyst in the liquid-phase hydrogenation of  nitrobenzene. Its look like extension work of “Russian Journal of Applied Chemistry, 2021, 94 (2), 223–229”. However, there are still several points found in the
manuscript requiring further elucidations after my reading in depth.

1.     Nickel Catalysts on Carbon-Mineral Sapropel-Based Supports for Liquid-Phase Hydrogenation of Nitrobenzene, I consider that there are other proposals better in the literature????

2.     The novelty of the work should be established. And its compared Russian Journal of Applied Chemistry, 2021, 94 (2), 223

3.     Figure 4: Provide EDS distribution in color form. Printing of the magazine is in color if it is distinguished but in black and white it is very difficult to visualize the differences.

4.     As for catalytic activity, this result is compared with the specific literature, include in your comparison results with other catalysts in tabular form.

5.     The introduction section was enriched by addition of the suggested literature regarding hydrogenation. Russian Journal of Applied Chemistry, 2021, 94 (2), 223; Catalysts 2022, 12(7), 767; Nanomaterials, 2022, 12(19), 3414; Materials Science for Energy Technologies 2022, 5, 391; Catalysts 202111(10), 1156.

Author Response

Reviewer 2

The manuscript entitled “Nickel Catalysts on Carbon-Mineral Sapropel-Based Supports for Liquid-Phase Hydrogenation of Nitrobenzene” describes Preparation of nickel catalysts with the carbon-mineral supports derived from sapropel and has been used as catalyst in the liquid-phase hydrogenation of  nitrobenzene. Its look like extension work of “Russian Journal of Applied Chemistry, 2021, 94 (2), 223–229”. However, there are still several points found in the manuscript requiring further elucidations after my reading in depth.

Our general response:

The authors thank the Reviewer for a careful analysis of the presented material. We tried to take into account all the recommendations. All changes are highlighted in yellow.

  1. Nickel Catalysts on Carbon-Mineral Sapropel-Based Supports for Liquid-Phase Hydrogenation of Nitrobenzene, I consider that there are other proposals better in the literature?

Thank you for bringing up this issue. Indeed, liquid phase nitrobenzene hydrogenation is a well-known reaction and many efficient catalytic compositions are used to carry it out. Some of them are given as an example in Table S1. As follows from the table, palladium is often used as an active component and expensive synthetic oxide and carbon materials as supports.

Often, this reaction (as in this work) is used as a test to compare the hydrogenation activity of a number of catalysts when optimizing their composition or synthesis conditions. In this case, mild conditions (low reaction temperature, short reaction time) should be used to exclude the influence of other factors, primarily deactivation. That is why in our work we carried out the reaction for 1 hour so that complete conversion was not achieved and we had a chance to correctly compare the catalysts and reveal the role of the support precursor and the active component precursor in the formation of nickel hydrogenation centers.

When testing catalysts based on sapropels under conditions close to industrial, as we have shown in the process of hydroliquefaction (Fuel. 2023, 332, 126300:1-10. DOI: 10.1016 /j.fuel.2022.126300), the activity of such catalysts is comparable to the activity of catalysts on other well-known supports.

The supports studied in this work are natural composites and contain oxide and carbon components, the ratio between which can be varied using different types of sapropel or special chemical treatments. The presence of carbon contributes to the development of specific surface due to micropores, and the mineral matrix provides high strength to the material, which is important in the creation of catalysts.

  1. The novelty of the work should be established. And its compared Russian Journal of Applied Chemistry, 2021, 94 (2), 223

We regret that we could not clearly show the novelty and originality of our work.

The main results of our research related to the synthesis and study of catalysts based on sapropel are present in [23, 26, 28]. These studies have established the influence of the nature of the original sapropel (ratios between organic and mineral parts) on the properties of the carbon-mineral material obtained from it, as well as the important role of the acid treatment of the support, which makes it possible to increase the specific surface area and the concentration of functional oxygen-containing groups. This modification led not only to an improvement in the cracking properties of catalysts with respect to large organic molecules [23, 28], but also to a decrease in the size of supported nickel particles [23, 26]. Another effective tool that made it possible to increase the dispersion of the active metal was the transition to the bimetallic (Ni–Mo and Ni–Cu) systems. With the use of such catalysts, with a decrease in the amount of nickel, an increase in activity was observed during the hydroconversion of the complex organic substance of sapropel to obtain bio-oil [28]. It is important that the properties of catalysts supported on carbon-mineral materials from sapropel were comparable to those of nickel-containing catalysts using more expensive synthetic supports such as alumina, amorphous aluminosilicate and ultrastable zeolite.

In this work, we considered the possibility of controlling the hydrogenating activity of nickel without using an additional metal, but by varying the nature of the active component precursor. Also in this study, the issue of the stability of catalysts based on sapropel and the possibility of their regeneration was considered; multi-cycle experiments were performed with the control of the nickel content in the catalysts.

To obtain supports, we also used sapropels from other deposits in order to show that the developed approaches to the synthesis and modification of composite carbon-mineral supports are applicable to any materials of this type, regardless of the raw material localization.

  1. Figure 4: Provide EDS distribution in color form. Printing of the magazine is in color if it is distinguished but in black and white it is very difficult to visualize the differences.

This is a really useful note. We have changed the color in Figures 4 and 6 (EDS data)

  1. As for catalytic activity, this result is compared with the specific literature, include in your comparison results with other catalysts in tabular form.

Thank you for your advice. We have added a comparison table of catalysts to supplementary materials (Table 1S). We will try to illustrate current trends: noble metal-based catalysts, noble-metal-free catalysts, catalysts containing synthetic supports and catalysts containing supports derived from natural raw materials.

Table S1. The comparison of the studied catalysts activity with other catalysts in the reaction of liquid-phase nitrobenzene hydrogenation

â„–

Catalyst

Reaction conditions

Conversion, %

Selectivity for aniline, %

Reference

1

Pd/carbon foam

P(H2)=2 MPa, T= 10-50oC, 5-240 min, methanol

99.1

99.8

Int. J. Mol. Sci. 202223(12), 6423;

https://doi.org/10.3390/ijms23126423

2

Pd/MnFe2O4

P(H2)= 2 MPa, 40-180 min, 10-50 oC, methanol

96

96

Int. J. Mol. Sci. 202223(12), 6535;

https://doi.org/10.3390/ijms23126535

3

Palladium-containing anion exchangers

P(H2) = 0.1 MPa, 20°C, ethanol

≥99

≥99

Petrol Chem, 2016, 56 (2), 146-150 DOI: 10.1134/S096554411602002X

4

Ru/MN270 (hypercrosslinked polystyrene)

P(H2)=0.2 МPа, 180 °C, 30 min,

 i-propanol

97

98

Bulletin of Science and Practice 2018, 4(12) http://doi.org/10.5281/zenodo.2254348

5

Ru/fulleride nanospheres C60

P(H2)=3 MPa, 80 °C, 240 min, ethanol

≥99

90

ACS Catal. 2016, 6, 6018−6024

DOI: 10.1021/acscatal.6b01429

6

Pd/carbon nanoglobules

P(H2)=0.5 МPа, 50 oC, 60 min, ethanol

≥99

≥99

Catalysis Letters 2020, 150 (3), 888-900

DOI: 10.1007/s10562-019-02974-6

7

Ni/TiO2

P(H2)= 1.96 MPa,

140 oC, 60 min,

ethanol

60

≥99

Chinese J Catal 2012, 33 (8),

DOI: 10.1016/S1872-2067(11)60398-7

8

Ni/Fullerene C60

P(H2)=1 МPа, 80 °C 20 min,

ethanol

99.5

≥99.9

Chemical Engineering Journal, 2020, 382 (15),122911.

https://doi.org/10.1016/j.cej.2019.122911

9

Ni/γ-Al2O3

P(H2)=4 – 8 MPa, 50 or 80 oC, 50 min Ethanol or scCO2

13-73

62-99

Journal of Catalysis 2009, 264 (1), 1-10.

https://doi.org/10.1016/j.jcat.2009.03.008

10

Ni/CM-M

P(H2)=2 MPa, T=90 oC, 60 min, ethanol

65

≥99

This work

  1. The introduction section was enriched by addition of the suggested literature regarding hydrogenation. Russian Journal of Applied Chemistry, 2021, 94 (2), 223; Catalysts 2022, 12(7), 767; Nanomaterials, 2022, 12(19), 3414; Materials Science for Energy Technologies 2022, 5, 391; Catalysts 2021, 11(10), 1156.

We are grateful for the provision of information on new interesting studies, and we used them in the "Introduction" section when describing nickel-containing catalytic systems using supports of various nature.

Reviewer 3 Report

Manuscript ID: catalysts-2105552

Title: Nickel Catalysts on Carbon-Mineral Sapropel-Based Supports for Liquid-Phase Hydrogenation of Nitrobenzene

The paper is interesting. It needs some rectifications before one can take a final decision:

1)   Novelty in comparison to previous works should be clarified within the text: https://doi.org/10.1134/S1070427221020129 ; http://dx.doi.org/10.1016/j.fuel.2022.126300.

2)   The paper contains some grammatical errors and typo-mistakes that should be corrected.

3)   The Abstract part should be improved. The abstract should clearly inform the important findings of the present study. Also, it should contain some qualitative and quantitative results.

4)   Change section “3. Materials and Methods” to be before “2. Results and Discussion”.

5)   A schematic illustration could be added, which will helpful for readers.

6)  The introduction part can be further improved. The authors are suggested to include recent references on similar subjects and discuss them very well in the introduction part.

7)  The structural properties should be further discussed and improved. Several structural parameters could be extracted from XRD patterns. Add hkl values to the XRD patterns.

8)  How are the “Textural characteristics of CM” determined? BET results should be provided.

9)  Along with TEM images, add the histograms of the particle size distribution.

10)     EDX spectra and atomic % should be provided.

11)     Kindly provide colored images for the elemental mappings.

12)     The conclusion should be more concise. It should also contain some qualitative and quantitative results.

Author Response

Reviewer 3

The paper is interesting. It needs some rectifications before one can take a final decision:

Our general response.

We are grateful to the reviewer for his/her positive overall evaluation of our article. We tried to take into account all the comments and, in accordance with them, made changes into the manuscript. All changes are highlighted in yellow.

  1. Novelty in comparison to previous works should be clarified within the text: https://doi.org/10.1134/S1070427221020129; http://dx.doi.org/10.1016/j.fuel.2022.126300.

The main results of our research related to the synthesis and study of catalysts based on sapropel are present in [23, 26, 28]. These studies have established the influence of the nature of the original sapropel (ratios between organic and mineral parts) on the properties of the carbon-mineral material obtained from it, as well as the important role of the acid treatment of the support, which makes it possible to increase the specific surface area and the concentration of functional oxygen-containing groups. This modification led not only to an improvement in the cracking properties of catalysts with respect to large organic molecules [23, 28], but also to a decrease in the size of supported nickel particles [23, 26]. Another effective tool that made it possible to increase the dispersion of the active metal was the transition to the bimetallic (Ni–Mo and Ni–Cu) systems. With the use of such catalysts, with a decrease in the amount of nickel, an increase in activity was observed during the hydroconversion of the complex organic substance of sapropel to obtain bio-oil [28]. It is important that the properties of catalysts supported on carbon-mineral materials from sapropel were comparable to those of nickel-containing catalysts using more expensive synthetic supports such as alumina, amorphous aluminosilicate and ultrastable zeolite.

In this work, we considered the possibility of controlling the hydrogenating activity of nickel without using an additional metal, but by varying the nature of the active component precursor. Also in this study, the issue of the stability of catalysts based on sapropel and the possibility of their regeneration was considered; multi-cycle experiments were performed with the control of the nickel content in the catalysts.

To obtain supports, we also used sapropels from other deposits in order to show that the developed approaches to the synthesis and modification of composite carbon-mineral supports are applicable to any materials of this type, regardless of the raw material localization.

  1. Terekhova, E.N.; Gulyaeva, T.I.; Trenikhin, M.V.; Muromtsev, I.V.; Nepomnyashchii, A.A.; Belskaya, O.B. Sapropel-based carbon mineral materials as catalyst supports for transformation of large organic molecules. Kinet Catal 2018, 59, 237-245.
  2. Terekhova, E.N.; Belskaya, O.B. Synthesis of nickel-containing sapropel based catalysts and their study in the liq-uid-phase hydrogenation of nitrobenzene. Russ J Appl Chem 2021, 94 (2), 223.
  3. Terekhova, E.N.; Belskaya, O.B.; Trenikhin, M.V.; Babenko, A.V.; Muromtzev, I.V.; Likholobov, V.A. Nickel catalysts based on carbon-mineral supports derived from sapropel for hydroliquefaction of sapropel organic matter. Fuel. 2023, 332, 126300:1-10.

  1. The paper contains some grammatical errors and typo-mistakes that should be corrected.

Thank you for your recommendation. We have tried to check the text carefully; typos that were found have been corrected.

  1. The Abstract part should be improved. The abstract should clearly inform the important findings of the present study. Also, it should contain some qualitative and quantitative results.

Thank you for your recommendation. Factual data was added to the Abstract and Conclusion

  1. Change section “3. Materials and Methods” to be before “2. Results and Discussion”.

Authors are recommended to use a Microsoft Word template for preparing the manuscript, which assumes just such an arrangement of parts of the article.

  1. A schematic illustration could be added, which will helpful for readers.

We have created a graphical abstract to display the content of the article. In accordance with your recommendations, we have also provided a support synthesis scheme (Fig. S9).

Figure S9. Scheme for the synthesis of CM support from sapropel.

  1. The introduction part can be further improved. The authors are suggested to include recent references on similar subjects and discuss them very well in the introduction part.

We appreciate the reviewer for this comment. We supplemented the literature review with new modern works related to the synthesis of nickel-containing catalysts on supports of various types, including biochars (refs. 2, 7, 10, 21). A brief analysis of the results of previous studies by the authors (refs. 23, 26, 28) was also performed to emphasize the novelty of the problems considered in this article. Comparative data are also presented for various types of catalytic systems used in the hydrogenation of nitrobenzene (Table S1).

  1. The structural properties should be further discussed and improved. Several structural parameters could be extracted from XRD patterns. Add hkl values to the XRD patterns.

Thank you for your advice. The main data on the structural properties of the catalysts, phase composition are presented in Fig. 2, Figs. S3, S4, S6 with links to PDF database. Some additional information on the composition, structure of supports based on sapropels and methods of its regulation can be found in the previous works of the authors.

1.Terekhova E.N.; Lavrenov A.V.; Shilova A.V.; Kireeva T.V.; Saveleva G.G.; Trenikhin M.V.; Belskaya O.B. Preparation of porous carbon–mineral materials by chemical treatment of sapropel carbonization products. Russ J Appl Chem 2017, 90, 1990-1997.

  1. Terekhova E.N.; Gulyaeva T.I.; Trenikhin M.V.; Muromtsev I.V.; Nepomnyashchii A.A.; Belskaya O.B. Sapropel-based carbon mineral materials as catalyst supports for transformation of large organic molecules. Kinet Catal 2018, 59, 237-245.
  2. Terekhova E.N.; Belskaya O.B. Cobalt-molybdenum catalysts based on carbon-mineral materials from sapropel for the large organic molecules transformation. AIP Conf Proc 2017, 1876, 020010:1-5.
  3. Terekhova E.N.; Belskaya O.B. Synthesis and study of bimetallic catalysts based on carbon-mineral materials derived from sapropel. AIP Conf Proc 2019, 2141, 020014:1-6.
  4. Terekhova E.N.; Belskaya O.B. Synthesis of nickel-containing sapropel based catalysts and their study in the liquid-phase hydrogenation of nitrobenzene. Russ J Appl Chem 2021, 94 (2), 223.
  5. Terekhova E.N.; Belskay, O.B.; Trenikhin M.V.; Babenko A.V.; Muromtzev I.V.; Likholobov V.A. Nickel catalysts based on carbon-mineral supports derived from sapropel for hydroliquefaction of sapropel organic matter. Fuel. 2023, 332, 126300:1-10.
  6. Krivonos O.I., Belskaya O.B. A New waste-free integrated approach for sapropel processing using supercritical fluid extraction. The Journal of Supercritical Fluids. 2020. V.166. 104991 :1-9.
  7. Krivonos O.I., Belskaya O.B., Likholobov V.A. Mechanical Activation as a Method to Regulate Morphology, Texture and Surface Functional Composition of Carbon-Mineral Materials Derived from Sapropel. Eurasian Chemico-Technological Journal. 2022. V.24. P.131-136.
  8. Terekhova E.N., Krivonos O.I., Belskaya O.B. Synthesis of Carbon-Containing Supports Based on Natural Raw Materials. Solid Fuel Chemistry. 2020. V.54. N6. P.373-384.

These works have shown that the presence of various compounds in a noticeable amount in a carbon-mineral supports makes it difficult to study in detail the structure of the particles of the applied active component. In this work, we were able to unambiguously identify the supported nickel and estimate (compare) particle sizes from the XRD data; (hkl) values for nickel are presented in fig. S3

  1. How are the “Textural characteristics of CM” determined? BET results should be provided.

Textural characteristics of the samples were investigated by the low-temperature nitrogen adsorption on a Sorptomatic 1900 Carlo Erba instrument (Italy). Prior to the adsorption measurements, the samples were evacuated at 300 oC for 6 hours to a residual pressure not higher than 0.1 Pa. Specific surface area of the samples was calculated according to BET from the adsorption isotherm in the range of equilibrium relative pressures of nitrogen Р/Р0 = 0.05–0.25. The micropore volume was estimated using a comparative t-method. The mesopore volume was calculated from the difference between pore volume at the equilibrium relative pressure of nitrogen vapor Р/Р0 = 0.990 and micropore volume. The macropore volume was found from the difference between pore volumes measured at P/P0 = 0.999 and P/P0 = 0.990. Details of adsorption measurements are present in the section 3, page 10. Quantitative data on the specific surface area as well as on the pore sizes are given in Table 2. On the example of samples tested in multicycle experiments, it was shown that the introduction of 10% nickel reduces the specific surface area from 160 to 100 m2·g-1.

  1. Along with TEM images, add the histograms of the particle size distribution.

Thank you for your recommendation. On fig. S5 histograms are presented for samples prepared using the same support CM-M but different nickel precursors, nitrate and formate. It follows from these data that a fairly uniform distribution is formed, and the average particle size for these catalysts is close (11.8–12.1 nm). The closeness of the sizes of supported nickel for these samples is consistent with the XRD data.

It should be noted that it is not always possible to obtain correct data on the particle size distribution, since the carbon-mineral support from sapropel contains high-density mineral components and it is difficult to identify particles of the introduced metal. In samples obtained from nickel sulfate, a new phase is formed, which also does not allow one to determine the size of nickel particles correctly from TEM data.

Figure S5. Size distribution of nickel particles from TEM data: a – Ni/CM-M, nitrate; b – Ni/CM-M, formate.

  1. EDX spectra and atomic % should be provided.

Thank you for your recommendation. On Fig. S1 the EDX spectrum and semi-quantitative analysis data for one of the sections of the “Ni/CM-M, formate” sample are shown, illustrating the chemical composition of the catalyst. Deviations in the values of local concentrations of deposited nickel from the results of chemical elemental analysis can be associated with both the inhomogeneous distribution of elements in the support based on natural raw materials and the influence of high concentrations of other components of carbon-mineral support.

element                                 C         O         Mg      Al        Si        S         Fe       Ni       

results in atomic%              23.86  45.52  0.35    1.03    14.83  2.39    1.13    10.88 

Figure S1. EDX spectra and atomic % of elements in sample Ni/CM-M, formate.

  1. Kindly provide colored images for the elemental mappings

Thank you for your advice. We have changed the color of the images (Figs.4 and 6)

  1. The conclusion should be more concise. It should also contain some qualitative and quantitative results.

Thank you for your recommendation. We have made changes to the Conclusion

Round 2

Reviewer 2 Report

Accept in present form

Reviewer 3 Report

The revised MS has been improved. I think that it can be now accepted for publication.